# Alien woody plants are more versatile than native, but both share similar therapeutic redundancy in South Africa

**Kowiyou Yessoufou**[1]*, **Annie Estelle Ambani**[1], **Hosam O. Elansary**[2], **Orou G. Gaoue**[1,3,4]

**1** Department of Geography, Environmental Management and Energy Studies, University of Johannesburg, Johannesburg, South Africa, **2** Plant Production Department, College of Food and Agriculture Sciences, King Saud University, Riyadh, Saudi Arabia, **3** Department of Ecology and Evolutionary Biology, University of Tennessee, Knoxville, TN, United States of America, **4** Faculty of Agronomy, University of Parakou, Parakou, Benin

\* kowiyouy@uj.ac.za

**Data Availability Statement:** All relevant data are within the manuscript and its Supporting Information files.

## Abstract

Understanding why alien plant species are incorporated into the medicinal flora in several local communities is central to invasion biology and ethnobiology. Theories suggest that alien plants are incorporated in local pharmacopoeias because they are more versatile or contribute unique secondary chemistry which make them less therapeutically redundant, or simply because they are locally more abundant than native species. However, a lack of a comprehensive test of these hypotheses limits our understanding of the dynamics of plants knowledge, use and potential implications for invasion. Here, we tested the predictions of several of these hypotheses using a unique dataset on the woody medicinal flora of southern Africa. We found that the size of a plant family predicts the number of medicinal plants in that family, a support for the non-random hypothesis of medicinal plant selection. However, we found no support for the diversification hypothesis: i) both alien and native plants were used in the treatment of similar diseases; ii) significantly more native species than alien contribute to disease treatments particularly of parasitic infections and obstetric-gynecological diseases, and iii) alien and native species share similar therapeutic redundancy. However, we found support for the versatility hypothesis, i.e., alien plants were more versatile than natives. These findings imply that, although alien plant species are not therapeutically unique, they do provide more uses than native plants (versatility), thus suggesting that they may not have been introduced primarily for therapeutic reasons. We call for similar studies to be carried out on alien herbaceous plants for a broader understanding of the integration of alien plants into the pharmacopoeias of the receiving communities.

## 1. Introduction

Elucidating what factors guide the selection of medicinal plants into a local pharmacopoeia is central to our understanding of human-plant interactions. Several alternative hypotheses have been formulated to explain the patterns of plant use by local people (see review in [1]).

**Funding:** This research was funded by King Saud University (Grant RSP-2020/118) and the National Research Foundation, South Africa (Grant 112113). KY is grateful to the South Africa's National Research Foundation (NRF) - Research Development Grants for Y-Rated Researchers (Grant No: 112113); OGG was supported by the Carnegie African Diaspora Fellowship and start-up funds from the University of Tennessee Knoxville. KY also received support from the University of Johannesburg in the form of a salary.

**Competing interests:** The authors have declared that no competing interests exist.

However, existing studies examined these hypotheses individually [2–4], and this limits our broad understanding of how multiple drivers shape local people's plant knowledge and use [5]. In the present study, we tested three interlinked hypotheses to explain the integration of alien plants into local medicinal flora: non-random selection, diversification, and versatility hypotheses. The link between these hypotheses can be justified as follows: humans decide to select a given alien plant to be introduced into a new environment (nonrandom selection hypothesis); once the species is introduced, the answers to the following questions determine its susceptibility to integrate local medicinal flora: Is the alien plant abundant or accessible to people (availability hypothesis)? Does it increase local medicinal plant species richness (diversification hypothesis)? Does it have additional uses, apart from medicinal uses, that may justify its selection and introduction to the new environment (versatility hypothesis)?.

The non-random plant selection hypothesis suggests that medicinal plants are not a random subset of total flora; rather, in its original form, the hypothesis predicts that the number of medicinal plants in a given family is predicted to increase linearly with the total number of species in the family [6]. It is now evident that the relationship between medicinal plants and family size is nonlinear [2]. In line with this hypothesis, family becomes a strong predictor of plant use value [7], and consequently, some families are over-utilized whilst others are under-utilized [6,8–10]. Over-utilized families are expected to be exceptionally rich in secondary compounds that are effective in the treatment of diseases [11,12], as opposed to under-utilized alkaloid-poor families [8,11,12].

In parallel, alien plants can also be introduced into a new geographic region originally to provide various services (e.g., food, construction materials, ornamental, etc.) and then later, their medicinal property may be realized, leading to their integration into local pharmacopoeia. Such scenario of multiple uses for the same plant species is predicted by the versatility hypothesis [5,12,13]. This inclusion of alien plants into local medicinal flora leads to an increase of the size of the latter, which, as a result, is now able to offer multiple treatment options to local communities. This increase (of the size of local flora and treatment options) is central to the prediction of the diversification hypothesis [14]. Evidence for the hypothesis has been reported in several studies [5,14–16].

All these hypotheses have been tested across different floristic and geographic regions, including America [6,10,11,17], Asia [18], South America [5,16,19–22], Europe [23], and southern Africa [24]. In southern Africa in particular, a floristically megadiverse region [25–29], we still have, however, a poor understanding of the theoretical basis for ethnobotanical patterns of human-plant interactions (but see [2]. The flora of South Africa, for example, represents 24,000 species, roughly 7% [26] of the world's 370,000 vascular plant species [12]. Of the South Africa's diversity, 2062 species are used and traded as medicinal plants [30], representing 10% of the total South Africa's flora [31] and 61% of all medicinal plants in the entire southern African region [32]. At the same time, several alien plants are introduced to the region for various purposes, including medicinal uses [33]. The lack of general understanding of the processes underlying the patterns of plant use in tropical Africa is due to the fact that most ethnobotanical studies in Africa aimed mostly to create a repertoire of medicinal plants used by local people (e.g. [34–36]).

In the present study, we used a multiple hypotheses approach to investigate the underlying processes of the patterns of plant selection for medicinal uses in southern Africa. Specifically, we first tested if the number of medicinal plants in a family is function of the size of that family (*non-random selection hypothesis*). Then, we identified families that are over- and under-utilized in the region. Finally, we investigated what drives the integration of alien plants into the regional medicinal flora (*versatility and diversification hypotheses*).

## 2. Materials and methods

### Ethics

Ethics approval (including permit application) and consent to participate to the study do not apply to this study. This is because the study required no field data collection since all data and information used in the study were retrieved from literature.

### Study area

The present study focused on southern Africa, a geographically delimited region comprising seven countries: Botswana, Lesotho, Mozambique, Namibia, South Africa, Swaziland and Zimbabwe. This region is well known for its megadiverse flora used for centuries by a diversity of peoples with different traditions and cultures for various purposes, including the medicinal one [37].

**The southern African woody flora.** Over a period of seven years (2008–2014), a team of the Botany Department from the University of Johannesburg (UJ), South Africa, has embarked in several botanical expeditions across the southern African region [38,39]. Through these expeditions, native trees and shrubs were identified and plant samples were collected for various herbaria including the UJ herbarium. Our definition of trees follows O'Brien [40]: trees are perennial plants that develop aboveground stem and secondary branches with a maximum height >2.5 m. Our definition of shrubs follows the one adopted in Bezeng et al. [41], i.e., species with a minimum height of 0.5 m. We also included few shrub species sometimes equivocally defined as herbaceous (e.g., *Tithonia spp.*, *Hypericum perforatum*) [42]. Collectively, we refer to our plant dataset simply as woody plant species.

Our woody dataset also includes alien species introduced into southern Africa. The checklist of alien species is informed by the latest checklist provided in Bezeng et al. [41] who combined the dataset of Henderson [43] with that of the Southern African Plant Invaders Atlas (SAPIA) [44] and Coates-Palgrave [25] with additional consultations with experts from the South African Biodiversity Institute and the Centre for Invasion Biology at Stellenbosch University, South Africa (see ref. [41]).

In total, the plant dataset used in the present study includes 1400 species (1190 native and 210 alien plants), representing 577 genera in 130 families: native (105 families), alien (46 families). All species names were cross-checked for synonyms using [45] and the taxonomy follows the Angiosperm Phylogeny Group IV [46].

**Medicinal status and various use categories of plants.** An extensive literature search was conducted to document the medicinal status (medicinal or non-medicinal) of all the 1400 plants in our dataset. The main source of information was the publications retrieved from the *Prelude Database for Medicinal Plants in Africa* [36], a database of most ethnobotanical studies in Africa, country by country, since 1847. All ethnobotanical information on the 54 African countries were regularly updated up to November 2017. From this unique database, we focused only on the selected countries in southern Africa and retrieved information of medicinal status of all species from the studies documented in this database. We additionally explored some other sources such as SANBIPlantZafrica [47] and (ethno)-botanical books that focused on southern Africa woody flora [25–29].

To document other plant use categories, first, we used Web of Science (WoS) to retrieve existing scientific ethnobotanical studies in the region. Second, we performed individual search for each species by using a combination of keywords such as "species name", "southern Africa", "Botswana", "Mozambique", "Namibia", "South Africa", "Swaziland", "Lesotho", "Zimbabwe", "uses", "usages", and "benefit". We also made use of Google and Google Scholar

using similar keywords to retrieve online resources such as regional and country-specific journals, proceedings, technical reports, herbarium, and commercial websites informing on the uses of woody plants in our dataset. SAPIA [44] was also consulted. In addition, we consulted key books on the regional flora (e.g. [25,33,48]). All the different uses retrieved from this wide and intensive literature search were grouped into 16 distinct use categories (S2 Table).

**Data on diversification and versatility.** Our diversification data included the following variables: i) different disease categories, ii) total number of species involved in the treatment of each disease category; and iii) species origin (native versus alien). Information on diseases treated by each of our species in the respective seven countries were retrieved from publications documented in the *Prelude Database for Medicinal Plants in Africa* [36] and the various literature indicated above. Some diseases were common or treated by the same species in more than one country in our study area. All diseases were grouped into 20 disease categories (S2 Table) based on the human body part concerned with the disease (e.g., kidney disease, etc.), and the data for species origin (native versus alien) were sourced as explained in the section "*The southern African woody flora*" above. All the different uses were grouped into 16 use categories (S1 Table).

## Data analysis

**Testing non-random medicinal plant selection.** To test if medicinal plants are non-random selection, we first determined the number of medicinal plants in each family and the total number of plants in the family. We then fitted a negative binomial model to the data using number of medicinal plants as response variable. The negative binomial model was preferred to the Poisson model to deal with overdispersion for a count data [49]. Finally, we identified over-utilized families as those with positive residuals (after fitting the negative binomial model); families with negative residuals were categorized as under-utilized. This analysis was done using the combined dataset (native + alien), and separately on native and then alien dataset.

**Testing diversification and versatility hypotheses.** To test the diversification hypothesis, we used two approaches. First, we calculated the number of native and alien species used in each disease category. Then we fitted the negative binomial generalized linear model (*glm.nb*) model to the data using the number of native and alien species treating a medical condition as response variable and disease category as predictor variable. To test if the species' origin (native or alien) made a significant contribution to explaining the number of species used for each disease category, we fitted an alternative *glm.nb* model including species origin as a co-variable. We selected the best model (between the model with and without species origin) using the Akaike Information Criterion (AIC) score of each model. The model with the lowest AIC score was selected as the best model after comparison of models (ΔAIC). If the best model includes origin, this implies that origin has a significant and unique contribution to the region's pharmacopoeia.

Second, we determined the medicinal redundancy score for each plant. This score was calculated for a given species as the average number of additional species recorded as treating the same disease with the given species (S3 Table). We then fitted a *glm* model to the redundancy score using total uses and plant origin (native vs. alien) as co-variables. A lower redundancy score for alien means that alien plants treat unique disease in the region.

In addition, we tested the relationship between the number of diseases treated by plants and their origins (S3 Table). This was done by fitting a negative binomial *glm* to the number of diseases. This model was fitted because our response variable is a count data and to avoid overdispersion. If alien species treated more diseases than native, this would imply that alien had unique contribution to regional pharmacopoeia.

Finally, to test the versatility hypothesis, we fitted a *poisson glm* to the total number of use categories using number of disease and origin as co-variables. From the data used, all non-medicinal plants are excluded (S3 Table). The *poisson glm* was used because the response variable is a count data and there was no overdispersion.

The R script used for the analysis is provided in Supplementary Information.

## 3. Results

### Nonrandom hypothesis

Most families of native and alien woody plants in southern Africa are medicinal (85% and 65%, respectively). In these families, the number of medicinal species increases with the size of the family: native flora ($\beta = 0.09 \pm 0.003$; $z = 25.1$, $p < 0.001$; Fig 1), and alien flora ($\beta = 0.05 \pm 0.007$; $z = 7.98$, $p < 0.001$; Fig 1), supporting the nonrandom hypothesis.

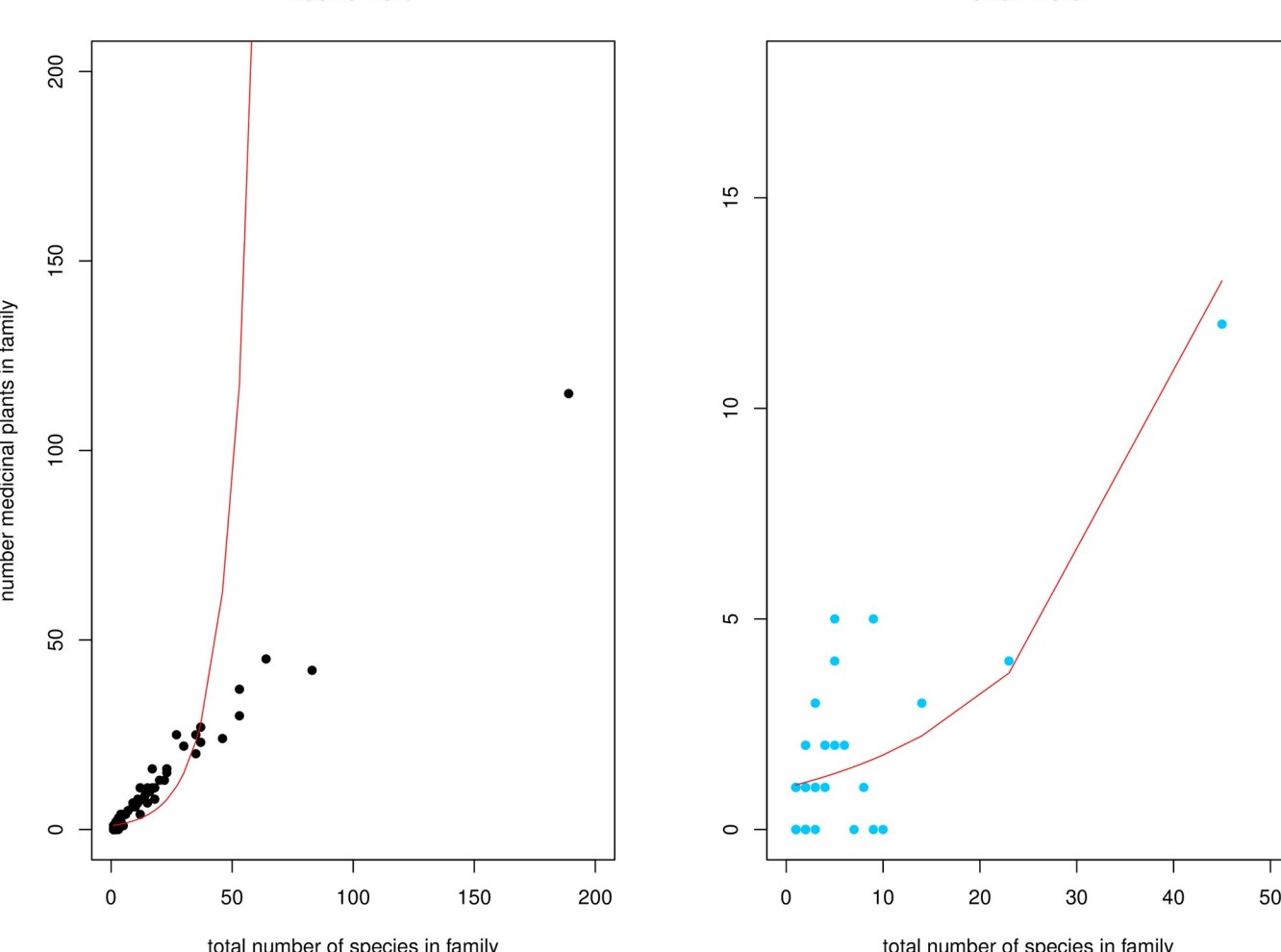

**Fig 1. Relationship between number of medicinal plant species and total number of species in native and alien flora of southern Africa.** The red line is the fit line following a negative binomial model. Each dot above the fit line represents an over-utilized family, and each dot below the red line represents an under-utilized family. Over-utilized family = family that contains more medicinal plants than expected, given the model fitted; Under-utilized family = family that contains less medicinal plants than expected.

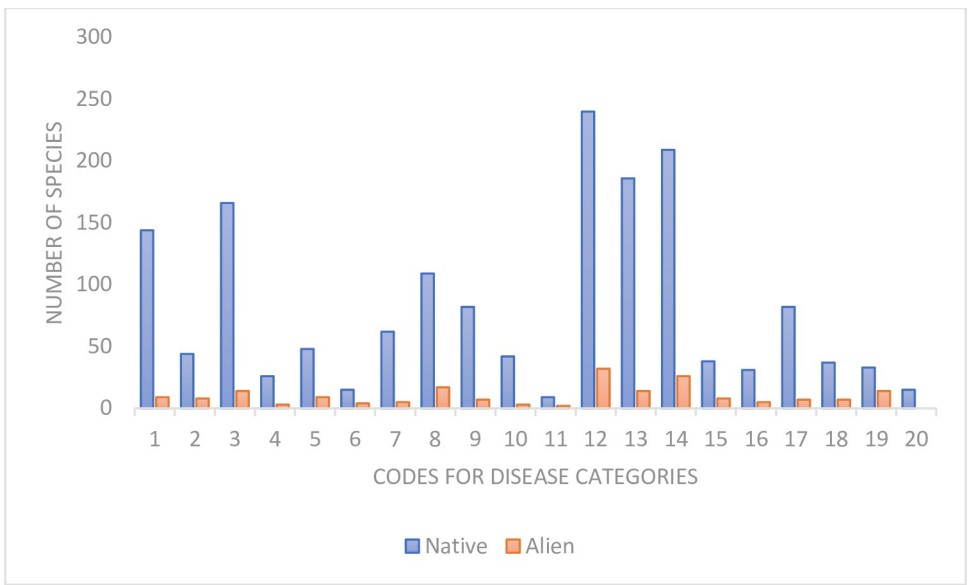

**Fig 2. Number of species used in the treatment of different diseases.** Disease categories are as follows: 1. Headache_Nervous_Mental_Disease, 2. Oral_Dental_Diseases, 3. Body_Pain_Killer_Anti_Inflammatory, 4. Kidney_Diseases, 5. Fever_Malaria; 6. Diabetes; 7. Burns_Wounds_Injury_Scars; 8. Respiratory_Diseases; 9. Skin_Diseases; 10. Eye disease; 11. Ear disease; 12. Parasitic_Infection_Control; 13. Gastrointestinal diseases; 14. Obstetrics_Gynaecology_Diseases; 15. cardiovascular diseases; 16. Growth disorders; 17. Blood disorder; 18. Anti_Venom_Poison; 19. HIV/AIDS; 20. Cancer.

An implication of this hypothesis is that some families are over-utilized and others are under-utilized. We found that the proportion of over-utilized families in the dataset of native species was ~ 51% (53 families), and the top 10 over-utilized families include Oleaceae (residuals = +1.62), Solanaceae (+1.62), Asphodelaceae (+1.57), Vitaceae (+1.30), Loganiaceae (+1.26), Rutaceae (+1.25), Thymelaeaceae (+1.082), Annonaceae (+1.080), Meliaceae (+1.07) and Apiaceae (+1.06) (Fig 1). It is important to highlight that the alkaloid-poor family Poaceae was also identified as over-utilized (+0.37) and the well-known highly threatened cycad family Zamiaceae was also part of over-utilized families (S4 Table).

The proportion of over-utilized families in the dataset of alien species was ~28%, and the top 10 over-utilized families included Euphorbiaceae (+2.31), Solanaceae (+1.96), Anacardiaceae (+1.78), Moraceae (+1.34), Rutaceae (+1.34), Adoxaceae (+0.72), Asparagaceae (+0.72), Apocynaceae (+0.59), Tamaricaceae (+0.59) and Bignoniaceae (+0.52). It is also important to highlight that the family Oleaceae, which was the top number one over-utilized family of native species now ranked last in the list of under-utilized families of alien species in all models (S4 Table).

## Diversification hypothesis

There is no support for the diversification hypothesis due to the following evidence: although we found a significant correlation between number of species used in treatments and diseases treated, the model which includes plant origin as co-variable outcompetes the model without origin ($\Delta_{AIC}$ = 31.5) but it is native species that showed significant contribution to treatments (Fig 2; β = 71.20 ± 14.27, t = 4.991, p<0.0001). In particular, native species contributes uniquely to the treatment of parasitic infections (β = 114 ± 45.11, t = 2.52, p = 0.02) and obstetric-gynecological diseases (β = 95.5 ± 45.11, t = 2.11, p = 0.04). For example, no alien plant was reported in the treatment of cancer whilst 15 native plants were used (Fig 2).

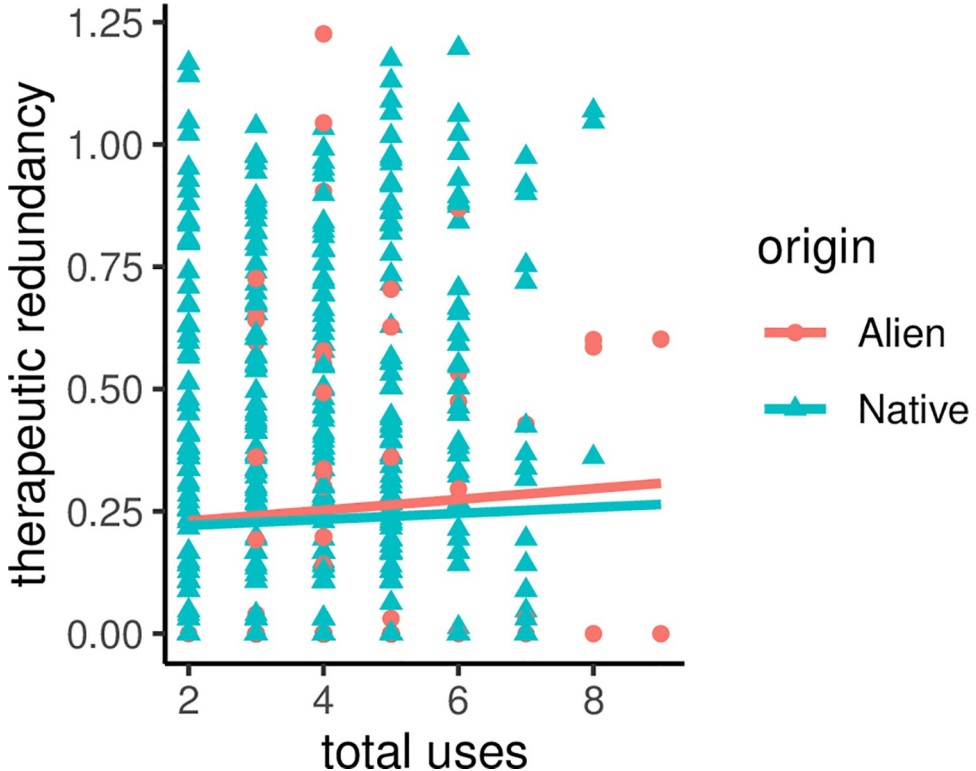

**Fig 3. Patterns of redundancy scores of alien vs. native plants.**

Even the test of redundancy also confirms no support for the diversification hypothesis. Specifically, although alien plant species had a slightly higher redundancy score and are used to treat a higher number of diseases than natives, this trend did not result in statistically significant difference (Fig 3): therapeutic redundancy ($\beta = -0.02 \pm 0.04$, $p = 0.596$) and number of diseases treated ($\beta = -0.14 \pm 0.11$, $p = 0.205$).

## Versatility hypothesis

Finally, there was support for versatility hypothesis such that alien plants have significantly more use categories than native plants ($\beta = -0.15 \pm 0.06$, $p = 0.008$; Fig 4).

## 4. Discussion

We found that most families of native and alien woody plants are used for medicine (85% and 65%, respectively); this is perhaps indicative of a vast richness of medicinal knowledge in southern Africa. Our analysis also indicated that this knowledge of medicinal plants is not, however, formed in a random manner. Instead, large plant families contain more medicinal plants than expected, thus supporting the non-random selection hypothesis of medicinal plants. Such positive relationship has been reported in numerous studies across different continents (Asia [18], South America [19,20], and North America [6,17,37,50], including in the Pacific [10], and in Africa [2,4]. Our results, using the southern African flora, provide additional support to the hypothesis, and we therefore suggest that the non-random hypothesis could perhaps be regarded as a generalizable pattern in ethnobotany.

As a consequence of this non-random selection, some families are over-utilized, that is, they contain more medicinal plants than expected, and this is a useful information to guide

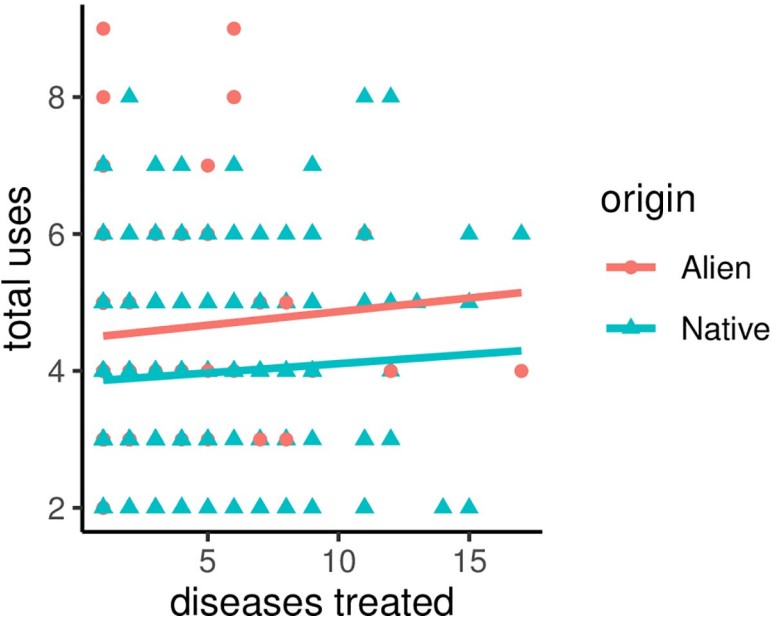

**Fig 4. Versatility patterns of alien vs. native plants.**

bio-prospection efforts for discovery of new medicinal plants. Indeed, the top 10 over-utilized families that we identified are variously rich in secondary compounds such as alkaloids, glycoside, and antioxidants [12], and the evidence that these families are widely used in the treatment of various diseases (e.g. fungal, parasitic, bacterial and microbial infections, hypertension and cardiovascular, gynecologic and obstetric problems, cancer and HIV/AIDS) was reported across different geographic regions [45,51–56], including southern Africa [24,35,47,48,57–66]. The convergent use of these families in different regions is perhaps an indication of their richness in secondary compounds, and such families may be targeted for bio-prospection [67]. There are, however, differences in the top 10 over-utilized families of native and alien species; these point perhaps to the evidence that alien plants did contribute to the diversification of local medicinal flora.

Surprisingly, our analysis also identified Poaceae as over-utilized woody family (e.g., *Oreobambos buchwaldii*, *Oxytenanthera abyssinica*, and *Thamnocalamus tesselatus*) in southern Africa (see also [10]). This is indeed surprising because the family Poaceae is an alkaloid-poor family, and its over-utilization in southern Africa could be a result of some unique traditional medicinal knowledge that needs to be further investigated in future studies. It could simply be a result of a historical cultural behavior transmitted across generations, but which is not grounded on any medicinal property. Elsewhere in the Pacific, the overutilization of Poaceae was attributed to its physical properties which make it a fast and dry material to burn and quickly apply for wound healing [68]. Over-utilization might also explain why some species have been listed as threatened in the region [69]. Our finding that the highly threatened family Zamiaceae [31,70] is among the over-utilized families is perhaps an elucidation that over-utilization may indeed lead to negative consequence on the future of some families [70].

Although 85% of native woody plant families are medicinal in southern Africa, we also noted that another 65% of the families of alien woody plants are incorporated in the regional pharmacopoeias. For example, the family Solanaceae of alien plants is over-utilized; so too is this family for native plants. This taxonomic overlap between native and alien species suggests Solanaceae might have high medicinal value in southern Africa's traditional medicine.

Interestingly, the Solanaceae family is also well known for its economic and nutritional values, and alien Solanaceae could potentially be introduced primarily for those values and, later on, be used for medicinal purposes as predicted by the versatility hypothesis [71,72]. For example, the historical introduction of alien woody plants to southern Africa was primarily motivated by the need to meet increased demands for charcoal, timber production, ornaments, and sand dune stabilization [73,74] and at a later stage, some of these introduced plants might have been used for medicinal purpose. Our test of the versatility hypothesis confirmed a significant difference in the number of use categories between native and alien plants, thus supporting our claim. However, native plants were more versatile than alien, thus providing an opportunity to further clarify why alien woody plants were introduced into the regional pharmacopoeia.

A potential alternative reason could be that alien woody plants are used medicinally simply because they are widespread across the geographic region as predicted by the availability hypothesis. Although we found that availability is a significant predictor of plant's medicinal status in our individual dataset of native and alien plants (see also [5,14,75]), the interaction between availability and plant origin (native versus alien) was not significant. This finding implies that, following Hart et al. [5], the availability of alien plants has no particular significance for their integration in the regional pharmacopoeia. This may point to the higher value attributed to the secondary chemistry or uniqueness of alien species relative to their sheer abundance. Alien species when they are abundance are likely to be invasive which can trigger negative perception from local people and limit their selection for medicine, unless they have unique therapeutic properties. We further investigated this claim by testing the diversification hypothesis in an attempt to explore whether alien plants introduced to southern Africa were actually unique in their contribution to the diversification of medicinal flora. To this end, three approaches were employed.

First, we tested if species' origin (native or alien) was a significant predictor of the number of plant species used for each disease category. We found that the model that includes plant origin was better than the model without plant origin. However, it was only native origin that showed significant correlation, confirming that alien contribution to medicinal use should be very minimal. The strict regulation and control of alien species introduction in the region (e.g., vast physical removal and destruction of "alien forests", see ref. [76,77] for alien control options in South Africa) may have contributed to limiting the availability of alien plants, and this could account for the non-significant interaction between plant origin and availability.

Second, we compared the redundancy scores of alien plants versus native plants. If alien plants were less redundant, this would mean that they were introduced to fill some therapeutic gaps in the regional pharmacopoeia. Our dataset *a priori* pointed to a possibility of taxonomic redundancy of alien plants since a very high proportion of alien families was shared with native plants (32 families representing 70% of alien families overlap with native families), suggesting that medicinal alien plants may not have a unique contribution to regional medicinal flora. An illustration of this is that the family Oleaceae is top over-utilized family in the native flora, but in the alien flora, it ranks last in the list of under-utilized families, indicating that, although alien Oleaceae has contributed to diversifying regional medicinal flora, it does not have significant contribution to medicinal purpose. Our analysis revealed indeed a trend towards more utilitarian redundancy for alien plants, albeit not significantly so.

Therapeutic redundancy hypothesis suggests that different species may provide similar medicinal uses to local people [78,79]. Redundancy has some benefits to therapeutically redundant species, which is that, in theory, they suffer less use-impacts due to the diffusion of use pressure among a wider array of plants [78]. However, recent evidence shows that less therapeutically redundant species may not suffer from a greater use pressure [3]. From ethnobotanic perspective, the implication of redundancy is that the loss of some redundant species

would not severely compromise local therapeutic practices [79]. However, redundancy may not necessarily result in less use-impacts for versatile species or when a redundant species is of cultural preference by local people [79,80]. In the present study, although alien plants show higher redundancy values than native plants, the difference is not statistically significant. This means that the proportion of alien plants that treat the same disease is similar to that of native plants. Alien plants are not unique in the study area; they are used to treat the same disease native plants treat except for cancer for which no alien plant is recorded. Further studies are necessary to tell the scenario (high or light use-pressure) that applies to the alien woody species included in the present study. Such studies may also include alien herbaceous species which may not necessarily show the same redundancy as the woody species since the latter may not primarily be introduced in the first place for medicinal purpose.

Finally, we also compared native and alien plants in term of the number of disease treatments they are involved in. Again, alien plants tend to treat more diseases than native, but the difference between native and alien was, once more, not significant. One explanation is that communities use common and available plants in their therapeutic practices irrespective of these plant origins. Also, dealing with diverse cultures in southern Africa, the selection or use of woody plants in the region might therefore differ [81] due to people's preferences [3]. This difference may blur the signature of a particular hypothesis in the context of a large scale ethnobotanic study [5].

Collectively, our multiple tests indicated that medicinal plant selection was not a random behaviour in southern Africa, and that alien plants are more versatility than native plants with which they share similar redundancy. This implies that alien plants were introduced to provide a wider range of use options, the benefit of which could be an increase in the resilience of regional medicinal flora in the face of environmental changes [22]. The lack of strong difference between alien and native in term of therapeutic redundancy may have to do with the scale of the study. Similar lack of evidence was also reported in a recent study conducted at national scale [22]. As pointed out in [22], this lack of evidence could stem from the differences in plant preferences in disease treatments due to differences in preferred biochemical pathways [82] or cultural differences such that the diseases treated by native plants in one environment or communities may be treated preferably by alien plants in another ones. The ethnobotanical force of people's preference was recently revealed in a study that showed that an increased preference decreases therapeutic redundancy [3]. This opens new windows for further studies at lower scale if we are to comprehend how medicinal plants are selected at village or community level. Future studies are also expected to test the therapeutic redundancy of alien herbaceous species which are not taken into consideration in the present study. We predicted a different outcome from that reported in the present study on woody plants since the latter may not primarily be introduced into South Africa for medicinal purpose unlike the herbaceous plants.

## Supporting information

**S1 Table. Use categories per species and origin.**
(XLS)

**S2 Table. Disease categories for alien and native.**
(XLS)

**S3 Table. Data on redundancy and versatility.**
(XLS)

**S4 Table. Residual values for each plant family.**
(XLS)

**S1 Rscript.**
(R)

## Acknowledgments

The authors extend their deep appreciation to King Saud University, Researchers Supporting Project (RSP-2021/118) and the University of Johannesburg for supports.

## Author Contributions

**Conceptualization:** Kowiyou Yessoufou.

**Data curation:** Annie Estelle Ambani.

**Formal analysis:** Kowiyou Yessoufou, Annie Estelle Ambani, Orou G. Gaoue.

**Funding acquisition:** Kowiyou Yessoufou, Hosam O. Elansary.

**Investigation:** Kowiyou Yessoufou.

**Methodology:** Kowiyou Yessoufou, Orou G. Gaoue.

**Project administration:** Kowiyou Yessoufou, Hosam O. Elansary.

**Supervision:** Kowiyou Yessoufou.

**Validation:** Kowiyou Yessoufou.

**Writing – original draft:** Annie Estelle Ambani.

**Writing – review & editing:** Kowiyou Yessoufou, Hosam O. Elansary, Orou G. Gaoue.

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
