## [Decision Letter · Decision Letter 0]

28 Sep 2020

PONE-D-20-22274

Therapeutic redundancy of alien medicinal woody plants in the southern Africa’s regional pharmacopoeia

PLOS ONE

Dear Dr. Yessoufou,

Thank you for submitting your manuscript to PLOS ONE. After careful consideration, we feel that it has merit but does not fully meet PLOS ONE’s publication criteria as it currently stands. Therefore, we invite you to submit a revised version of the manuscript that addresses the points raised during the review process.

Thank you for submitting you paper to Plos One. The reviewers made a god and thoughtful job. Please, pay attention to all the comments, especially regarding to clarify all the hypothesis put under evaluation in your manuscript.

We look forward to receiving your revised manuscript.

Kind regards,

Prof. Dr. Ulysses Paulino Albuquerque

Academic Editor

PLOS ONE

Journal Requirements:

2.Thank you for stating the following in the Acknowledgments Section of your manuscript:

[AEA (ORCID ID 0000-0003-2040-681X) acknowledges

470 funding from the DST-NRF Innovation Doctoral Scholarships (Grant # 98719)

471 and a PhD merit bursary offered by the University of Johannesburg. KY is

472 grateful to the South Africa’s National Research Foundation (NRF) - Research

473 Development Grants for Y-Rated Researchers (Grant No: 112113); OGG was

474 supported by the Carnegie African Diaspora Fellowship and start-up funds

475 from the University of Tennessee Knoxville.]

 [This research was funded by King Saud University, grant number RSP-2020/118 and the National Research Foundation, South Africa, grant number 112113.]

3.We note that [Figure(s) 1] in your submission contain [map/satellite] images which may be copyrighted. All PLOS content is published under the Creative Commons Attribution License (CC BY 4.0), which means that the manuscript, images, and Supporting Information files will be freely available online, and any third party is permitted to access, download, copy, distribute, and use these materials in any way, even commercially, with proper attribution. For these reasons, we cannot publish previously copyrighted maps or satellite images created using proprietary data, such as Google software (Google Maps, Street View, and Earth). For more information, see our copyright guidelines: http://journals.plos.org/plosone/s/licenses-and-copyright.

1.    You may seek permission from the original copyright holder of Figure(s) [1] to publish the content specifically under the CC BY 4.0 license. 

Reviewers' comments:

Reviewer's Responses to Questions

**Comments to the Author**

1. Is the manuscript technically sound, and do the data support the conclusions?

Reviewer #1: Partly

Reviewer #2: Partly

Reviewer #3: Partly

2. Has the statistical analysis been performed appropriately and rigorously? 

Reviewer #1: N/A

Reviewer #2: Yes

Reviewer #3: Yes

3. Have the authors made all data underlying the findings in their manuscript fully available?

Reviewer #1: Yes

Reviewer #2: Yes

Reviewer #3: No

4. Is the manuscript presented in an intelligible fashion and written in standard English?

Reviewer #1: Yes

Reviewer #2: Yes

Reviewer #3: Yes

5. Review Comments to the Author

Reviewer #1: The paper aims to test four important hypotheses of ethnobiology (non-random hypothesis, availability hypothesis, versatility hypothesis and diversification hypothesis) using data from woody medicinal plants from southeastern Africa. The publication of this study can be of great contribution to ethnobiology, mainly because it makes a broad analysis about the entry of exotic species in local pharmacopoeias. However, I suggest a major revision before the manuscript is considered for publication. I hope the authors consider trying to clarify some ideas. Below are my comments and suggestions for change.

First, I felt a disconnection between the title and the ideas presented and discussed. The title makes us think that the work is about the redundancy of exotic species, but in fact it is about a greater scope of hypotheses that can explain such redundancy or the absence of it, among other things in the selection of medicinal plants. Redundancy and its implications, after all, is little discussed.

Secondly, I think the hypotheses could be better presented in the background. The diversification hypothesis needs to be better defined, as it seems to be confused with the versatility hypothesis. In addition, there is a lack of dialogue between the non-random hypothesis and the other hypotheses that deal with the entry of exotic species in local pharmacopoeias. Another point about this hypothesis is that, according to what I understand from the non-random hypothesis (and I may be mistaken, so the need to explain it better in the background), the number of medicinal plants in a given family is predicted to NOT increase linearly with the total number of species in the family, precisely because some families are over-utilized whilst others are under-utilized.

In the background the authors define the ways to measure availability, however, in the methodology, they apparently do not use any of the metrics presented to test this hypothesis. Instead, they redefine availability as the total number of countries where the plant occurs. I believe that a plant can occur in many countries but not necessarily be of high availability in those locations. I hope that the authors clarify this measurement, otherwise it would be necessary to rethink whether the study can really propose to test such a hypothesis.

Regarding the test of the diversification hypothesis, the measure for redundancy is very interesting, but I was thinking that a low redundancy score can still reveal an overlap between native and exotic (for example, a disease treated only by two plants, presenting low redundancy, however including a native and an exotic plant). I suggest, in addition to this analysis, to compare the number of medicinal categories in which there are only exotic with those that exist both or only native, which would show that the exotic entered the medical system to fill the gaps (diversification hypothesis). On the third test for diversification, exposed in lines 255-258, I believe it can show how the exotic are more important in the medical system than native or vice versa, but it is not a measure of diversification. One of the consequences of the diversification hypothesis is that exotic species can first enter the medical systems to fill blanks (diversification hypothesis) and then their application can be extended to the other diseases already treated by the native plants, and this would result in a greater number of medicinal categories. In other words, this test is extremely important for the discussion about the unfolding of the diversification hypothesis, and it should present a good discussion, but it is not in itself a measure of diversification.

In general, the paper brings very interesting results, which perhaps deserve to be discussed further, presenting some possible hypotheses for future studies (such as the surprising highlight of the Poaceae family). I believe that many ideas reached the conclusion without being previously discussed, such as the idea that exotic species did not enter to diversify, but to increase redundancy, which would contribute to resilience (I suggest reading Nascimento A.L.B. et al. (2015)* if the authors believe redundancy is an interesting idea to explain their results).

Finally, I think it is important to add a section presenting the limitations of the study. In addition to the observations noted above, any ethnobotanical study based on other studies, whose data collection was carried out by different researchers, probably using different methodologies and rapport efforts, has limitations, which does not invalidate the results (lines 166-167: "a database of all ethnobotanical studies that ever took 167 place in Africa, country by country, since 1847 "). In addition, I think it is important to emphasize the importance of herbaceous species for traditional pharmacopoeias, and that if they were included in the analyzes, perhaps the results would be different.

*Utilitarian Redundancy: Conceptualization and Potential Applications in Ethnobiological Research. In: Albuquerque U., De Medeiros P., Casas A. (eds) Evolutionary Ethnobiology. Springer, Cham. (2015) https://doi.org/10.1007/978-3-319-19917-7_9

Reviewer #2: The study intitled “Therapeutic redundancy of alien medicinal woody plants in the southern Africa’s region pharmacopoeia” contributes to the evaluation of hypotheses that seek to explain the selection of exotic species in different pharmacopeias on a regional scale. However, different aspects throughout the text need to be made clearer. Below, I highlight these points.

-In the introduction, the first sentence indicates “factors guide the selection of medicinal plants". Right after, the paragraph indicates “patterns of plant use” and “drivers shape local people plant knowledge and use". In this sense, I suggest to indicate more clearly that the hypotheses in the paragraph are related to the selection of medicinal plants. In addition, at the end of this paragraph, the following passage is indicated: "However, existing studies examined these hypotheses individually and this limits our global understanding of how multiple drivers shape local people plant knowledge and use [2]." However, the reference “2” contributes to assess theses hypotheses jointly, so it is necessary to highlight what remains to be done, even considering the study “2”.

-At the beginning of the second paragraph, I suggest deleting the following passage to make the text more objective: "rather, the number of medicinal plants in a given family is predicted to increase linearly with the total number of species in the family".

-From line 84, the following passage is indicated: “In parallel, alien plants can also be introduced into a new geographic region primarily for diverse services, including food, construction materials, ornamental, etc. and then be used for medicinal purpose when the need to fill some therapeutic gaps in local pharmacopoeias arises." Next, it is stated that such a scenario involves the versatility hypothesis. However, the last passage: “…and then be used for medicinal purpose when the need to fill some therapeutic gaps in local pharmacopoeias arises" is not necessarily related to the versatility hypothesis, but rather to the diversification hypothesis. The diversification hypothesis in fact indicates that exotic plants are incorporated to fill local therapeutic gaps (Albuquerque 2006, mentioned in the manuscript) and this is not clearly indicated in the work of Bennett and Prance (2000, also mentioned in the manuscript). Although both hypotheses are indicated in the paragraph, it would be interesting to adjust the excerpts in which the hypotheses are presented for that the reader is not confused.

-In line 105, I suggest removing the term “theories”, because so far in the introduction, hypotheses have been presented.

-In the topic “Medicinal status and various use categories of plants”, it would be interesting to clarify whether inclusion and exclusion criteria were used to select documents in the search for literature on the uses of species. In this case, did the documents obtained undergo a methodological quality assessment? Sometimes, many papers in a region may interview a few people and this may lead to the failure to register certain plants that would have medicinal use in a community or region, which would have implications for the results of this research.

-In line 211, it would be interesting to specify whether any criteria were used to organize diseases into 20 different categories.

-In line 231, I suggest changing “dependent variables” to “predictor” or “independent variables”.

-In line 272, would (53 species) actually be (53 families)?

-In line 280, I suggest starting the paragraph by presenting the proportion of over-utilized families for exotic species, just as it was presented for native species.

-In line 313, it is stated that “there was a support for versatility hypothesis such that native plants tend to have more use categories than alien". According to the argument presented in the introduction to the versatility hypotheses, wouldn’t it be the opposite result (exotic species being indicated for a greater number of categories than native) that would support the hypothesis? I have this same question in the discussion topic, on lines 365-370. Thus, I believe that the text can be adjusted.

-In the discussion, there is an indication that over-utilized families are rich in secondary compounds, except the Poaceae family, which is an alkaloid-poor family. It would also be interesting to present in the results the under-utilized families (both for native and exotic species) and indicate in the discussion whether these families tend to have low richness of secondary compounds. In addition, there are differences in the 10 over-utilized families of native and exotic species. It would also be the case to present a discussion on these findings.

-In line 377, I suggest replacing “(native versus native)” to (native versus exotic).

-At the end of the discussion, I missed an explanation about the findings indicating non-significant differences between the redundancy values and the number of treatments between native and exotic species. In this case, in the last paragraph of the discussion, the following excerpt is indicated: “Also, dealing with diverse cultures in southern Africa, the selection or use of woody plants in the region might therefore differ...". It would be interesting to develop a little more that explanation. In the topic of the conclusion, there are moments when possible explanations are presented, considering scenarios (such as the resilience scenario) and evidence presented in the literature. So, I suggest expanding these explanations in the discussion topic and summarizing the information present in the conclusion.

Reviewer #3: In this paper, the authors propose to explore the drivers of medicinal uses of woody plants in the Southern African region (including seven countries). They do so by testing four hypotheses about plant selection and use (non-random, availability, versatility and diversification). The large datasets used for the study are built from literature reviews (for plant uses, including medicinal) and available botanical datasets.

The hypotheses that are being tested are quite clear, the methodology is well described. The analyses that are proposed are conducted correctly - despite some issues with data, see below for details. The main results obtained are i) to show that medicinal plant use in this area is non-random, thus confirming the non-random hypothesis; ii) to identify some possibly over-used families of plants, which open ways for further research, and iii) to show that the three other hypotheses didn’t explain in a significant way the variations observed.

It is an interesting paper, covering a large area; there has been a lot of work involved in literature review and collection of data into coherent datasets.

There is, however, a number of issues that need to be addressed by the authors to make their paper more convincing, regarding the analyses, the datasets, the wording of conceptual elements, and the limits of the study, particularly in relation with the chosen scale of analysis.

Major comments

1. The authors should provide more justification about the choice of the study area: why selecting these seven countries, and not limiting the study to one country, or to one bioclimatic region, which would eliminate some flaws related to the large-scale, while keeping a dataset large enough for statistical analyses?

2. Many dimensions likely to account for plant use are not taken into account in this study (let’s just mention the socio-cultural background). This should be mentioned in a dedicated paragraph, as well as other limits to the study that need to be spelled out clearly.

3. In the same line, while the methods to constitute the datasets are well explained, and show that this task has been carried out with a lot of attention, it is important to also acknowledge the potential limits of these datasets. For example, the sources used to list medicinal uses are likely to reflect only a part of the actual uses, or to be based on individual accounts only (as is sometimes the case with early ethnobotanical studies), thus reflecting only a part of what is happening “on the ground”.

About the non-random hypothesis

4. The way to explain and refer to the non-random hypothesis and associated results in the paper could be clearer. To my understanding, there are two main “components” in this hypothesis, as it is found in the literature.

The first component is that there can be a trend, to be observed in the data, that the number of species with medicinal uses increases in relation with family size. This relation can be linear, logarithmic, and else. However, there is a key point here: if the relation is strictly linear, then it means there is no differences between families, the proportion of medicinal plants being the same across families. It shows that the odds of a species to be medicinal is constant across families, which points towards randomness rather than non-randomness. The second component is then to show, with the help of residuals or any other method, the differences across families, some hosting a higher number of medicinal plants than expected, some a lower. The first component alone is not sufficient; as used by Moerman 1979, it is interesting insofar as it allows to look at the residuals, at the differences between families, and the residuals only will “prove” differences between families (and the removal of families and genera with a strong deviation then lead to a better correlation coefficient - idem). The same point is underlined by Moerman & Estabrook 2003: medicinal uses are related to families - regardless of the size of the families.

So I recommend rephrasing or nuancing the sentences dealing with this hypothesis, as showing that size of families is a predictor of the number of medicinal plants it hosts is not directly a confirmation of the non-random hypothesis “in full” (as differences between families are not revealed this way): abstract (l. 36-38), l.65-69, 112-113, and in the discussion and conclusion (l.321-322: the results don’t show that all large families contain more medicinal plants than expected, only some of them).

5. Many families included in the sample only host one species (out of 127 families, 41 are represented by only 1 species), and many families display zero values for medicinal plants.

The binomial regression models may be affected by a high number of zero counts (cf Zuur et al. 2010): I suggest to provide details about this aspect of the dataset and provide justification about the validity of the results.

(Zuur et al. 2010. A protocol for data exploration to avoid common statistical problems. Methods in Ecology and Evolution (1):3-14.)

About the versatility hypothesis

6. While the versatility hypothesis is introduced in the paper in relation with medicinal plants, the analysis seems to have been conducted with all plant species, including the ones that don’t have any medicinal use - no details are provided (l.260-262, and results 313-315). It would be useful to i) precise if only plants with at least one medicinal use were taken into account, and ii) if not, repeat this analysis with a subset including only plants with a documented medicinal use (826 out of 1400 according to table S2), to keep the analysis in line with the hypothesis it is aiming to test. The results and discussion sections related to this analysis should be changed accordingly.

About the datasets

There are discrepancies in the datasets, that need to be corrected, as well as the analyses depending on these datasets:

7. Table S1: some species are recorded as non-medicinal (col.4) while having medicinal uses counted (col.6): Ekebergia_pterophylla_OM3263 in Meliaceae, several eucalyptus species in Myrtaceae. The residuals in table S4 are probably wrong for these families.

8. Conversely, many species are recorded as medicinal in S1, col.4, but without any precise medicinal use counted in col.6 (408 species in total). The redundancy score is null for all these species. Given that among these 408, 30 are alien and 378 native, it introduces an important bias in the results: redundancy scores are skewed towards zero among the native species. The glm models and W tests testing the diversification hypothesis should be repeated, excluding the species with missing values - another solution would be to complete the data, which I understand can be challenging if the literature is not precise enough. In all cases the authors should mention clearly if they worked with a subset or the whole dataset, why, and how it affects the overall results of their study. The results and discussion sections related to this analysis should be changed accordingly.

Statistics

9. It would be useful to the readers to have a summary table providing the basic descriptive statistics of the variables used in the analyses (e.g. number of medicinal plants per family, average number of medicinal uses per species,...), in a table inserted in the text or in supplementary material. This would allow the reader to get a quick grasp on the structure of the datasets and on variables’ value distribution.

10. In the result section, some basic descriptive analyses could strenghten the key results, and show that exploratory analyses have been conducted. I would suggest to present these first, then move to inferential analyses such as the binomial models. For example, for the non-random hypothesis, the correlation between family size and number of medicinal species should be tested, and results provided and eventually discussed (there is a strong - almost perfect - linear correlation in the case of native flora), before moving to binomial model results.

11. In the same line,in the case of diversification, the results of the Wilcoxon tests should be presented first, before the model results (l.298-311).

12. A similar test should be done in the case of the availability hypothesis (fig.3, l.289-296).

Other comments

13. Regarding the phrasing of the epistemological relation between the hypotheses and the facts they are dedicated to help understand (i.e. plant uses and plant-related knowledge): some sentences should be nuanced to acknowledge the fact that the hypotheses that are being tested in the paper are very general and cannot account by themselves alone for the complexity of human-plant interactions (as can be described e.g. in an ethnographical work). I therefore recommend rephrasing the sentences l. 54-55 and 97-98.

14. Over-utilization as revealed in the results (i.e. as an output of binomial regression models) doesn’t account for the quantity of plants that are being harvested in the environment. So the results point towards a possible over-exploitation, but do not prove it in any way (a species can have many uses but being harvested in small quantities). I recommend nuancing the way this is expressed - e.g. l. 347-351.

15. The authors mention several botanical surveys carried out between 2008 and 2014 (l.138-148). However we don’t know to what extent these surveys contributed to the dataset. How many species or presence of species/families were documented and added during these? Would the datasets have been uncomplete without these? Moreover, if these surveys provided an important part of the botanical datasets, how has data on uses of these species been collected, in relation with the places they were collected in? If some species were previously unknown in some of these areas, one can expect a lack of data on uses and knowledge about them.

16. The title doesn’t reflect appropriately the content of the article.

17. Following PlosOne data policy, a table providing the details of medicinal uses per species should be made available to the readers.

Minor comments:

18. The reference 46 doesn’t seem to be the right one to be cited l.159.

19. The table S.2 counts 826 species with a medicinal use, the S.1 displays 825.

20. The families Monimiaceae and Stangeriaceae don’t appear in the table S4, while being included in S1.

21. It would be useful to add a column with the results of the combined native + alien negative binomial model (for non-random hyp.) to the table S4.

22. L.349-351: the residuals for the Zamiaceae is quite low (0,077, the lowest positive residuals value in the analysis), I wouldn’t interpret it as showing over-utilization.

23. The reference to fig.3, l.292 is not adapted: the figure doesn’t illustrate results of the binomial model.

6. PLOS authors have the option to publish the peer review history of their article (what does this mean?). If published, this will include your full peer review and any attached files.

Reviewer #1: No

Reviewer #2: No

Reviewer #3: **Yes: **Matthieu Salpeteur

---

## [Author Response · Author response to Decision Letter 0]

22 Oct 2021

Johannesburg 22nd October 2021

Cover Letter

Dear Editor,

First of all, we thank you for keeping the online resubmission open after such a long time despite your repeated requests to resubmit the paper. The Covid-19 pandemic has disorganized the normal way of academic life, and this explains that. 

Second, we have now revised the ms following closely the comments of each of the 3 reviewers, and the response letter is provided below. We also provide a revised ms with track changes so that you can follow all our changes.

We have added the following statement to the caption of Figure 1: “The shapefile to generate Fig 1 was obtained from https://www.diva-gis.org/Data”

We also agree that the funding statement below appears alongside the published paper:

Funding: This research was funded by King Saud University (Grant RSP-2020/118) and the National Research Foundation, South Africa (Grant 112113). KY is grateful to the South Africa’s National Research Foundation (NRF) - Research Development Grants for Y-Rated Researchers (Grant No: 112113); OGG was supported by the Carnegie African Diaspora Fellowship and start-up funds from the University of Tennessee Knoxville. KY also received support from the University of Johannesburg in the form of a salary.

We thank you again,

The Authors

---

## [Editor Report · Decision Letter 1]

4 Nov 2021

PONE-D-20-22274R1Alien woody plants are more versatile than native, but both share similar therapeutic redundancy in South AfricaPLOS ONE

Dear Dr. Yessoufou,

Thank you for submitting your manuscript to PLOS ONE. After careful consideration, we feel that it has merit but does not fully meet PLOS ONE’s publication criteria as it currently stands. Therefore, we invite you to submit a revised version of the manuscript that addresses the points raised during the review process. ==============================

The authors submitted a much improved version of their work. I request some additional changes to make reading the text more fluent: Please restructure the results to make a clear and objective allusion to whether a given hypothesis was confirmed or refuted, and then present the results of the analyses.

 Please submit your revised manuscript by Dec 19 2021 11:59PM. If you will need more time than this to complete your revisions, please reply to this message or contact the journal office at plosone@plos.org. Please include the following items when submitting your revised manuscript:A rebuttal letter that responds to each point raised by the academic editor and reviewer(s). You should upload this letter as a separate file labeled 'Response to Reviewers'.A marked-up copy of your manuscript that highlights changes made to the original version. You should upload this as a separate file labeled 'Revised Manuscript with Track Changes'.An unmarked version of your revised paper without tracked changes. You should upload this as a separate file labeled 'Manuscript'.If applicable, we recommend that you deposit your laboratory protocols in protocols.io to enhance the reproducibility of your results. Protocols.io assigns your protocol its own identifier (DOI) so that it can be cited independently in the future. For instructions see: https://journals.plos.org/plosone/s/submission-guidelines#loc-laboratory-protocols. Additionally, PLOS ONE offers an option for publishing peer-reviewed Lab Protocol articles, which describe protocols hosted on protocols.io. Read more information on sharing protocols at https://plos.org/protocols?utm_medium=editorial-email&utm_source=authorletters&utm_campaign=protocols.

We look forward to receiving your revised manuscript.

Kind regards,

Ulysses Paulino Albuquerque

Academic Editor

PLOS ONE
---

## [Editor Report · Decision Letter 2]

10 Nov 2021

Alien woody plants are more versatile than native, but both share similar therapeutic redundancy in South Africa

PONE-D-20-22274R2

Dear Dr. Yessoufou,

We’re pleased to inform you that your manuscript has been judged scientifically suitable for publication and will be formally accepted for publication once it meets all outstanding technical requirements.

Kind regards,

Ulysses Paulino Albuquerque

Academic Editor

PLOS ONE
---

## [Editor Report · Acceptance letter]

18 Nov 2021

PONE-D-20-22274R2 

Alien woody plants are more versatile than native, but both share similar therapeutic redundancy in South Africa 

Dear Dr. Yessoufou:

I'm pleased to inform you that your manuscript has been deemed suitable for publication in PLOS ONE. Congratulations! Your manuscript is now with our production department. 

Kind regards, 

on behalf of

Dr. Ulysses Paulino Albuquerque 

Academic Editor

PLOS ONE